# No-Regret and Incentive-Compatible Combinatorial Online Prediction

## Abstract

We study the combinatorial online learning prediction problem with bandit feedback in a strategic setting, where the algorithm selects a subset of experts at each round and the experts can strategically influence the learning algorithm's predictions by manipulating their beliefs about a sequence of binary events. There are two learning objectives for the algorithm. The first is maximizing its cumulative utility over a fixed time horizon, equivalent to minimizing regret. The second objective is to ensure incentive compatibility, guaranteeing that each expert's optimal strategy is to report their true beliefs about the outcomes of each event. In real applications, the learning algorithm only receives the utility corresponding to their chosen experts, which is referred to as the full-bandit setting. In this work, we present an efficient algorithm based on mirror descent, which achieves a regret of $O(T^{3/4})$ under both the full-bandit or semi-bandit feedback model, while ensuring incentive compatibility. To our best knowledge, this is the first algorithm that can simultaneously achieve sublinear regret and exact incentive compatibility. To demonstrate the effectiveness of our algorithm, we conduct extensive empirical evaluation with the algorithm on a synthetic dataset.

## 1 Introduction

We study the combinatorial online prediction problem, in which a learning algorithm is asked to make predictions about a sequence of $T$ binary events. This is the extension of the classic online learning with expert problems (Vovk, 1990; Littlestone & Warmuth, 1994; Cesa-Bianchi et al., 1997; Kivinen & Warmuth, 1999; Chen et al., 2013). At each round, the learning algorithm selects $m \geq 1$ experts out of $k$ experts, each has a belief about the likelihood of the event. Based on the chosen expert, the algorithm will then receive a utility. The goal of the algorithm is to select a sequence of subsets of experts that is as accurate as possible. Usually, the performance is then measured in regret, which is the difference between the cumulative utility received by the algorithm and the best cumulative possible with the best subset of experts' predictions possible. The class of algorithms that attain sublinear regret are referred to as the no-regret algorithms.

In real applications, however, the experts may often be strategic individuals. They may strategically manipulate their predictions to maximize their own utility. For instance, the renowned website FiveThirtyEight aggregates data from various pollsters to predict election outcomes and evaluates them by publicly scoring their accuracy and methodology. The pollsters may be incentivized to tailor their methodologies or reporting practices to align with what they believe will garner higher scores, to gain influence. More motivating examples can be found in the appendix.

Under the manipulation of strategic experts, it is desirable to have the algorithm be incentive-compatible, for two reasons. First, when the experts strategically skew their reported prediction, the resulting aggregate prediction may drift away from accuracy, leading to erroneous conclusions and decisions. Second, incentivizing experts to be honest could reveal more truth to the public, which is beneficial to society. However, many of the classic online learning algorithms, such as the multiplicative weight, are shown to be not incentive-compatible (Freeman et al., 2020). This thus motivates the design of incentive combative algorithms that are also no-regret.

In the special case of $m = 1$, Freeman et al. (2020) presented the first incentive-compatible and no-regret learning algorithm. The algorithm is based on the wagering mechanism, which is a type

of multi-agent scoring rule and is known to be incentive-compatible in the offline setting. The algorithm is shown to enjoy sublinear regret under both full information (where the algorithm receives information about the accuracy of every expert) and the bandit setting (where the algorithm only receives information about the accuracy of the chosen experts). Follow-up work by Zimmert & Marinov (2024) then improves the regret bound to $O(\sqrt{T})$ under the bandit setting.

Sadeghi & Fazel (2023) is the first work to study the general case, where $m \geq 1$. They presented two algorithms, one is based on the perturbed follow-the-leader algorithm, and the other one is based on the greedy algorithm, which attains the $O(\sqrt{T})$ regret in the full information setting. However, their results are restricted to the case where the performance of the experts is evaluated with a quadratic loss, and the utility of the algorithm takes a specific form. Moreover, the algorithm only guarantees approximate incentive compatibility, or when the utility is a specific modular function. It thus remains open whether there is an algorithm that is no-regret and incentive-compatible under bandit feedback, and general utility function.

In this paper, we present the first algorithm that is no-regret and incentive-compatible for the combinatorial online prediction problem under bandit (and semi-bandit) feedback and monotone utility function feedback. We show the equivalence between the online mirror descent and modified linear update rule, in which case we can simultaneously gain the no-regret property from the mirror descent update and the incentive compatibility from the linearized update rule. To overcome the challenge of the full bandit feedback, we introduce a one-sample gradient estimator over a hypercube. We then conclude the $O(T^{3/4})$ regret upper bound by utilizing the continuous relaxation of the original objective. We complement our theoretical findings on a synthetic dataset, in which our algorithm outperforms the online mirror descent algorithm in both regret minimization and the incentive compatibility guarantee.

## 2 RELATED WORKS

A long line of works have explored the connection between online learning and strategic predictions, as well as strategic behaviors in multi-armed bandits (Abernethy & Frongillo, 2011; Frongillo et al., 2012; Liu & Chen, 2016; Braverman et al., 2019; Feng et al., 2020; Dong et al., 2022; Witkowski et al., 2023). The closest setting to ours is Roughgarden & Schrijvers (2017); Freeman et al. (2020); Zimmert & Marinov (2024); Sadeghi & Fazel (2023). In Roughgarden & Schrijvers (2017), the experts are assumed to be selfish and their objective is to maximize the unnormalized parameter weights of the learner algorithm. In this case, it has been shown that the multiplicative weight algorithm is sufficient to ensure incentive compatibility while maintaining low regret. In reality, the experts are often more interested in maximizing the probability of being chosen, which is considered by Freeman et al. (2020). In the proposed setting, there is a proper loss function is drawn from an unknown distribution $\mu_t$ at every round and there are $k$ experts who each have a private belief $b_t$ about the distribution but report a prediction $p_t$. The goal of the experts is to maximize their chances of being selected. Freeman et al. (2020) show that the classic online learning algorithms, such as the online mirror descent, are not immediately incentive-compatible. To simultaneously attain sublinear regret and incentive compatibility, Freeman et al. (2020) presented the weighted score update (WSU), which is based on the wagering mechanism, a type of multi-agent scoring rule. They then show that the algorithm enjoys a regret of $O(\sqrt{T})$ for both the full information (where the algorithm receives all the loss information) and a regret of $O(T^{2/3})$ the partial information setting (where the algorithm only receives the loss information corresponds to the expert chosen) while being incentive-compatible. Follow-up work by Zimmert & Marinov (2024) then showed that the WSU can be shown as a first-order approximation linear update of the classic exponential weight algorithm. By adjusting the update rule to minimize the approximation error, they presented an improved regret of $O(\sqrt{T})$ in the bandit setting.

The above-mentioned work still differs from our setting, as the learning algorithm only chooses one expert at each round and commits to their prediction. Sadeghi & Fazel (2023) extended the online incentive-compatible learning problem to the combinatorial setting, where the learning algorithm is required to select a subset $S_t$ at each round, which contains $m \geq 1$ experts. They assumed that the loss is quadratic, and the utility is function either $\frac{|S_t|}{m} - \frac{1}{m} \sum_{i \in S_t} \ell_{t,i}$, which is modular, or $1 - \prod_{i \in S_t} \ell_{t,i}$, which is submodular. They represented a followed perturbed leader

algorithm that achieves $O(\sqrt{T})$ regret under full information, but it is only approximately incentive-compatible. They then proposed a distorted greedy algorithm that achieves a better regret and is exactly incentive-compatible when the utility function is modular. However, the algorithm requires computing the marginal increase in utility of including any expert, which makes it inapplicable to the bandit feedback setting. We summarize the difference between the two works in the following table.

Table 1: A comparison of the setting considered by the previous work (Sadeghi & Fazel, 2023) and our work.

| | Loss | Utility Function | Feedback | Incentive Compatibility | Approximation Factor |
|---|---|---|---|---|---|
| Prvious Work | Quadratic | Specific Known Function | Full Information | Approximate | $1 - \frac{c_f}{e}$ |
| **Ours** | Proper | Any Unknown Submodular Function | Full Bandit | Exact | $1/2$ |

## 3 PRELIMINARIES

We consider the incentive-compatible $(k, m)$-experts problem described by Sadeghi & Fazel (2023), which is a combinatorial extension of the incentive-compatible multi-armed bandits problem studied by Freeman et al. (2020); Zimmert & Marinov (2024). There are $k$ experts available in this problem, and each expert makes a prediction about a sequence of $T$ binary outcomes. At time step $t \in [T]$, each expert $i \in [k]$ holds a private belief $b_{t,i} \in [0, 1]$ about outcome $r_t \in \{0, 1\}$. Each expert $i$ then reports a prediction $p_{t,i}$ to the learner. Based on the reported prediction, the learner chooses a subset $S_t$ of no more than $m$ experts, i.e. $|S_t| \leq m$. Then, the outcome $r_t$ is revealed and each expert induces a loss $\ell_{t,i} = \ell(p_{t,i}, r_t)$, where $\ell : [0, 1] \times \{0, 1\} \to [0, 1]$ is a proper loss function, which is formally defined in Definition 3.1 and includes a wide range of the popular loss functions such as the quadratic loss. The utility of the learner at each round $t$ is $f(S_t, r_t)$. We allow the private beliefs $\{b_{t,i}\}_{i=1}^k$ and outcome $r_t$ to be chosen arbitrarily and potentially adversarially.

**Definition 3.1.** *A loss function $\ell$ is said to be proper if*

$$\mathbb{E}_{r \sim \text{Bern}(b)}[\ell(p, r)] \geq \mathbb{E}_{r \sim \text{Bern}(b)}[\ell(b, r)], \quad \forall p \neq b.$$

If a loss function is proper and the outcome indeed follows the same Bernoulli distribution as the expert's private belief, then the expert will induce a lower loss if they report honestly. We further maintain the following assumptions about the utility function $f$.

**Assumption 3.1.** *For any $i \in [k]$ and any $S \in \mathcal{S}, r$, where $\mathcal{S}$ is the set of possible combinations of less than $m$ experts, $f(S, r)$ is a monotone submodular function that is linear and with negative derivative with respect to each $\ell_{t,i}$.*

We assume that the experts are self-interested with the goal of maximizing the probability of being selected at every round. That is $p_{t,i} = \arg\max_p \mathbb{E}_{t,i}[x_{t+1,i} \mid p_{t,i} = p]$, where $x_{t+1,i}$ is the probability of expert $i$ being selected at round $t + 1$.

**Learning Objectives** There are two objectives for the learner:

1. To maximize the cumulative utility as much as possible;

2. To incentivize the experts to report their private beliefs truthfully,

Formally, the first objective can be achieved by minimizing the $\alpha$-Regret, which is defined as

$$\alpha - \text{Regret}(T) = \mathbb{E}\left[\alpha \max_{S \subseteq [K]:|S| \leq m} \sum_{t=1}^{T} f_t(S) - \sum_{t=1}^{T} f_t(S_t)\right].$$

In the offline version of the problem, $\alpha = 1 - c_f/e$, where $c_f \in [0, 1]$ is the curvature of the sub-modular function, which is shown to be optimal (Sviridenko et al., 2017). While the previous work of Sadeghi & Fazel (2023) gives a regret with the optimal approximation factor, their algorithm is not immediately incentive-compatible unless the utility function satisfies $c_f = 0$. As our second objective is to satisfy the incentive compatibility completely, we consider the regret with an approximation factor of $1/2$. It remains open whether the approximation factor is tight with the additional constraint of incentive compatibility.

In a setting where the experts can be strategic, incentive compatibility can be crucial for the regret to be meaningful. This is because the regret definition measures the performance of the hindsight best subset of experts, which can be bad given the strategic behaviors of the experts. We thus consider the second objective, the incentive compatibility. If an algorithm is incentive-compatible, it would then be in the best interest of any expert $i$ to truthfully report $p_{t,i} = b_{t,i}$. The following definition describes an incentive-compatible online learning algorithm.

**Definition 3.2.** *An online learning algorithm is incentive-compatible if for every $t \in [T]$, every expert $i \in [k]$ with belief $b_{t,i}$, every report $p_{t,i}$, reports of other experts $p_{t,-i}$, every history of reports $\{p_{s,j}\}_{j \in [k], s < t}$, and outcomes $\{r_s\}_{s < t}$, we have:*

$$\mathbb{E}_{r_t \sim \text{Bern}(b_{t,i})} \left[ x_{t+1,i} \mid b_{t,i}, p_{t,-i}, \{p_{s,j}\}_{j \in [K], s < t}, \{r_s\}_{s < t} \right]$$

$$\geq \mathbb{E}_{r_t \sim \text{Bern}(b_{t,i})} \left[ x_{t+1,i} \mid p_{t,i}, p_{t,-i}, \{p_{s,j}\}_{j \in [K], s < t}, \{r_s\}_{s < t} \right],$$

*where Bern(b) denotes a Bernoulli distribution with probability of success $b$ and $x_{t+1,i}$ is the probability of expert $i$ being chosen at round $t + 1$.*

**Full Bandit Feedback and Semi-Bandit Feedback**  Different from Sadeghi & Fazel (2023), which assumes full access to each $\ell_{t,i}, \forall i \in [k], t \in [T]$ and the structure of utility function $f$ for designing no-regret algorithm with incentive compatibility, we consider the $m$-experts problem under either full or semi bandit feedback.

- Full bandit feedback: In this case, the learner only receives $f(S_t, r_t)$ upon committing the action of choosing a subset of experts $S_t$. Notice that the learner does not have access to any loss $\ell_{t,i}$ incurred by an individual expert $i$ at any time step $t$.
- Semi-bandit feedback: In this case, the learner has access to the losses $\ell_{t,i}, \forall t \in [T]$ if $i \in S_t$. Notice that the learner does not have access to $\ell_{t,i}$ for $i \notin S_t, \forall t \in [T]$.

## 4 ALGORITHM FOR FULL BANDIT FEEDBACK

In this section, we introduce our algorithm with only full bandit feedback. Our approach is based on a continuous relaxation of the original problem, which allows us to perform gradient-based updates that are demonstrated to be incentive-compatible. Subsequently, we utilize an extension mapping to derive a discrete solution $S_t$ for each time step.

---

**Algorithm 1:** Algorithm for Full Bandit Feedback

**Input:** Learning rate $\eta_t$, parameter $\delta_t$, Safety hypercube $\mathbb{H}_r(p) \subseteq \mathcal{X}$

1 **for** $t = 1, \ldots, T$ **do**
2     Sample $z_t$ sampled from $\{\pm 1\}^k$.
3     Update $\omega_t = z_t - r^{-1}(x_t - p)$.
4     Update $\tilde{x}_t = x_t + \delta_t \omega_t$.
5     Sample $S_t$ from $\text{EXT}(\tilde{x}_t)$
6     Receive $f(S_t, r_t)$ and set $g_t = \frac{-k f(S_t, r_t) z_t}{2 \delta_t}$
7     Set $\tilde{g}_{t,i} = g_{t,i} + \frac{k}{2\delta_t}$
8     Update $x_{t+1,i} = x_{t,i} \left( 1 - 2\eta_t \sqrt{x_{t,i}} \tilde{g}_{t,i} \right)$

---

## 4.1 CONTINUOUS RELAXATION

In order to employ gradient-based optimization on $f$, we want to relax the problem of optimizing $f$ on $\mathcal{S}$, the set of possible combinations of less than $m$ experts on a continuous function $F$. One of the most intuitive choices is to define $F$ as the multilinear extension of $f$: let $F(x, r) : \mathcal{X} \to \mathbb{R}$ be the multilinear extension of $f(S, r)$, i.e. $F(x, r) = \sum_{S \in \mathcal{S}} \prod_{i \in S} x_i \prod_{j \notin S} (1 - x_j) f(S, r)$.

Then, if we could estimate the function value of the multi-linear extension unbiasedly by using only one query to the corresponding discrete submodular function, we can reduce the problem of optimizing $f$ over a discrete set $\mathcal{S}$, to optimizing $F$ over $\mathcal{X} \subseteq [0, 1]^k$. However, this is generally impossible to achieve in the full bandit feedback setting, as the definition of the multi-linear extension relies on information about the values of the submodular set function across all subsets, including those that do not meet the specified size constraints (Zhang et al., 2019). A formal proof of impossibility is given in Lemma 2 of Zhang et al. (2019). To overcome this, Zhang et al. (2019) considered a relaxed setting, where the algorithm is allowed to select an infeasible subset and receive zero reward. Under this setting, they designed an algorithm to attain $O(T^{8/9})$ regret. Although it is infeasible to find a one query estimator for the general case, we remark that this does not rule out the possibility of one query estimator for specific loss and constraint set.

Instead of following the relaxed setting, we consider a class of problems with the appropriate extension $F$. Specifically, we make the following assumption on $f$.

**Assumption 4.1.** *For a finite set $\mathcal{S}$ that includes possible subsets containing fewer than $m$ items from a total of $k$ items, a binary value $r$, and a monotone, submodular function $f$, there exist an extension mapping $EXT : \mathcal{X} \to \mathcal{S}$, $\mathcal{X} = \{x : x \in [0, 1]^k, \|x\|_1 \leq m\}$ such that*

1. *$F(x, r) = \mathbb{E}_{S \sim EXT(x)}[f(S, r)]$, is a multilinear, monotone, DR-submodular function, and $F$ is $L$-Lipschitz continuous.*

2. *For any $S \in \mathcal{S}$, there exists an $x$ such that $EXT(x) = \mathbb{I}_S$, where $\mathbb{I}_S$ assign probability $1$ to $S$ and $0$ to other elements of $\mathcal{S}$.*

We can thus reduce the problem of maximizing $f$ over $\mathcal{S}$ to maximizing $F$ over $\mathcal{X}$. A similar assumption was also made in bandit submodular maximization literature (Wan et al., 2023), which is shown to be satisfied for several important functions $f$ that have applications in fields such as online retailing platforms optimization. Moreover, such extension mapping is shown to exist for partition matroid constraint (Lemma 5.5 in Wan et al. 2023). As the cardinality constraint we considered is a special case of partition matroid constant, Assumption 4.1 is indeed satisfied for our case.

## 4.2 ONE-POINT GRADIENT ESTIMATION

To perform gradient-based optimization on the continuous relaxation $F(x)$, we would need to build a gradient estimator with just the feedback $f(S_t, r_t)$. To achieve this, we leverage the idea of a one-point gradient estimator in the bandit optimization literature (Flaxman et al., 2005). In the context of bandit optimization, where the action domain resides within the probability simplex, a gradient estimate of a function $h(x)$ can be derived from a single function evaluation by selecting a random unit vector $u$ and scaling it by evaluating $h(x + \delta u)$, where $\delta$ is a tunable parameter. This approach is appropriate because the expectation of the gradient thus obtained is proportional to the gradient of $h$ smoothed over the surface of a unit sphere.

We extend this idea to our continuous relaxation $F$, and build the gradient estimation error by randomly smoothing $F$ over a hypercube of side length $\delta_t$. Specifically, given $\delta_t \geq 0$, define $\tilde{F}(x, r) = \mathbb{E}_z [F(x + \delta_t z, r)]$. We can show that

$$\mathbb{E}_{\partial[-\delta_t, +\delta_t]^k} [F(x + \delta_t z, r) z] = \frac{2\delta_t}{k} \nabla \tilde{F}(x, r).$$

Then $g_t = \frac{-k f(S_t, r_t) z_t}{2\delta_t}$ is an unbiased gradient estimate of $-\tilde{F}(x_t, r_t)$, when $S_t$ is a subset formed according to $\tilde{x}_t = x_t + \delta_t z_t$. Since $F$ is smooth, the bias of our estimator can be controlled by tuning $\delta_t$, $\|\nabla \tilde{F}(x_t, r_t) - \nabla F(x_t, r_t)\| \leq O(\delta_t)$.

However, one more challenge remains as we have a cardinality constraint of $\|x + \delta_t z_t\|_1 \leq m$. A natural idea may be to choose a suitable $\delta_t$ such that the smoothing ensures the $\tilde{x}_t$ still remains

feasible. Yet it is unclear how to adjust $\delta_t$. We thus choose to do a feasibility adjustment of $z_t$ based on a *safety hypercube*, instead of directly adjusting $\delta_t$. Let $\mathbb{H}_r(p)$ be a hypercube of side length $r$ such that $\mathbb{H}_r(p) \subseteq \mathcal{X}$. Then, instead of perturbing $x_t$ by $z_t$, we adjust it by computing

$$\omega_t = z_t - r^{-1}(x_t - p), \quad \tilde{x}_t = x_t + \delta_t \omega_t.$$

Then,

$$\tilde{x}_t = (1 - \delta_t r^{-1})x_t + \delta_t r^{-1}(p + 2z_t),$$

$\tilde{x}_t$ remains feasible if $\delta_t/2 < 1$, and we play $\tilde{x}_t$ instead. The bias of the gradient estimator is thus $2\delta_t L$ due to the $\delta_t$ amount of perturbation and the feasibility adjustment.

While there is no closed-form expression for $p$, it can be selected practically through empirical adjustments. Specifically, $p$ can be the center of the hypercube $\mathbb{H}_r(p)$ if: 1) Each coordinate of $p$, $p_i$, satisfies $p_i \in \left[\frac{r}{2}, 1 - \frac{r}{2}\right]$. 2) The $\ell_1$-norm constraint $\sum_{i=1}^{k} p_i \leq m - \frac{kr}{2}$ is met. To find a suitable $p$, we can start with an initial assignment of $p_i = \frac{m - \frac{kr}{2}}{k}$ for each $i$. If any $p_i$ falls outside the interval $\left[\frac{r}{2}, 1 - \frac{r}{2}\right]$, we iteratively adjust the values of $p_i$ to bring them within bounds while ensuring that the sum constraint remains satisfied.

Equipped with the gradient estimator, we now discuss how to perform with gradient-based optimization on $F$ while ensuring incentive compatibility.

### 4.3 Linear Update and Incentive Compatibility

A greedy algorithm such as the continuous greedy algorithm may be appealing to minimize the regret of optimizing a submodular function in an online setting. However, previous work has shown that such an approach is not immediately incentive-compatible and may only achieve approximate incentive compatibility in some specific settings, such as the loss is quadratic (Sadeghi & Fazel, 2023). On the other side, popular online algorithms such as online gradient descent and the exponential weight are also shown to be not incentive-compatible (Freeman et al., 2020).

One straightforward way to ensure incentive compatibility is to define $x_{t+1,i}$ as a linear function with a positive derivative for $f(S_t, r_t)$. Consequently, maximizing $x_{t+1,i}$ becomes equivalent to maximizing $f(S_t, r_t)$ in expectation. Given that $f(S_t, r_t)$ is assumed to be linear and with a negative derivative with respect to each $\ell_{t,i}$, expert $i$ will need to minimize $\ell_{t,i}$ in order to maximize $x_{t+1,i}$. Since the loss $\ell$ is assumed to be proper, it naturally follows from the definition of a proper loss function. This idea was previously used in deriving incentive compatibility bandit algorithms (Freeman et al., 2020; Zimmert & Marinov, 2024).

However, one challenge in selecting the linear update rule is ensuring that the updated $x_{t+1}$ still stays in $\mathcal{X}$, which has constraints $\|x\|_1 \leq m$ and $x_i \in [0,1], \forall t \in [T], i \in k]$, and to have sublinear regret. While classic algorithms such as online projected gradient descent or mirror descent can easily solve this, it is unclear how to modify a linear update rule to fulfill this. To address this challenge, we update $x_{t+1,i}$ with $x_{t+1,i} = x_{t,i}\left(1 - 2\eta_t\sqrt{x_{t,i}}\tilde{g}_{t,i}\right)$, where $\tilde{g}_{t,i}$ is a shifted one-point gradient estimator, $\tilde{g}_{t,i} = g_{t,i} + \frac{k}{2\delta_t}$. We demonstrate that this update is equivalent to the well-known online mirror descent method, with the regularizer defined as $\Phi(x) = -2\sum_{i=1}^{k}\sqrt{x_{t,i}} + \mathbb{I}_{\mathcal{X}}(x)$, $\mathcal{X} = \{x \in [0,1]^k : \sum_{i=1}^{k} x_i \leq m\}$. Notice that this choice of regularizer guarantees our update $x_{t+1}$ stays in $\mathcal{X}$, and the equivalence allows us to utilize the classic mirror descent analysis to desrive a regret guarantee. A similar equivalence between a modified online mirror descent and a linear-style update rule has been established when $x$ is restricted to the probability simplex (Zimmert & Marinov, 2024). We further extend the idea to the $m$-experts problem.

Overall, our algorithm is summarized in Algorithm 1.

## 5 Analysis for Full Bandit Feedback

In this section, we present the main theorems for the full bandit feedback setting with Algorithm 1.

**Theorem 5.1.** *With $\eta_t = \frac{1}{kt^{3/4}}, \delta_t = \frac{4}{t^{1/4}}$ and $f$ satisfying assumption 4.1, we have*

$$\mathbb{E}\left[\sum_{t=1}^{T} f(S^*) - 2f(S_t)\right] \le O\left(D_\Phi(x^*, x_1)k^{3/2}T^{3/4} + m^2 kT^{3/4} + mLT^{3/4} + kGT^{3/4}\right).$$

We first highlight the $\frac{1}{2}$ approximation factor and the order of our regret bound. For submodular optimization problems, a $(1 - c_f/e)$ approximation ratio is known to be optimal for any algorithm that makes polynomially many queries to the objective function, where $c_f$ denotes the curvature constant of the submodular function (Sessa et al., 2019). However, in the online setting, achieving an approximation ratio of $O(1-1/e)$ typically requires variants of the Frank-Wolfe algorithm (Chen et al., 2018; Zhang et al., 2019). The primary challenge with using the Frank-Wolfe algorithm in this context is the incentive compatibility constraint. The Frank-Wolfe algorithm involves an inner optimization step along the gradient direction, but it is unclear whether this step maintains incentive compatibility. In fact, when the constraint $m = k$, the Frank-Wolfe algorithm reduces to traditional gradient descent, which has been shown to be non-incentive-compatible (Freeman et al., 2020).

To satisfy the incentive-compatible constraint, we resort to the Follow-the-Regularized-Leader (FTRL) algorithm. To our knowledge, Wan et al. (2023) is the only work to achieve both a $(1-1/e)$ approximation ratio and a regret bound of $O(T^{3/4})$ using the FTRL algorithm under full bandit feedback. However, this approximation ratio was achieved through a delicate gradient transformation technique (Lemma 2.6 and Eq (5) in Wan et al. (2023)), for which it remains unclear whether the incentive compatibility constraint is satisfied. Finally, the $O(\sqrt{T})$ regret from non-incentive-compatible online combinatorial optimization (Audibert et al., 2014) is not directly comparable, as their analysis restricts the loss function to be linear.

**Theorem 5.2.** *Algorithm 1 is incentive-compatible.*

Under the full information setting, where the learner can obtain $\ell_{t,i}, \forall t, i$, Sadeghi & Fazel (2023) showed a regret bound of $O\left(\sqrt{T \ln(k/m)}\right)$, obtained with an distorted greedy algorithm. As their algorithm is a variant of the greedy algorithm, they require full information to compute the marginal increase on including each $i$. However, this result is limited to the case where $\ell$ is a quadratic loss and their algorithm is not fully incentive-compatible. To our best knowledge, Theorem 5.1 is the first regret bound achieved by an incentive-compatible algorithm under the full bandit feedback setting.

Different from the full information result, Theorem 5.1 suffers from a dependency of $k^{3/2}$ due to the gradient estimator. Since the gradient estimator serves as an unbiased approximation of the utility function smoothed over a hypercube, the resulting bound for gradient estimation inherently scales with the dimension of the hypercube. In the next section, we show that this dependency in $k$ can be improved and the result can be strengthened to a high probability one when we have access to semi-bandit feedback.

Compared to the distorted greedy algorithm proposed by Sadeghi & Fazel (2023), our algorithm gains computational efficiency as it only requires one sampling procedure to obtain the subset $S_t$ at each round. The distorted greedy algorithm can be seen as a meta-algorithm that runs $m$ instances of weighted score updates (WSU). To obtain a subset $S_t$, the algorithm samples each WSU algorithm for an expert. In the case where two WSU algorithms recommend the same expert, the algorithm will need to do a resample. Thus the computation complexity can be bad in the worst case.

While we defer the proofs to the appendix, we discuss the main ideas of the proof here.

*Proof Sketch.* With the continuous relaxation, we first upper bound $\sum_{t=1}^{T} F(x^*) - 2F(x_t)$. Using the smoothness, monotonicity, and submodularity of $F$, we can decompose it as

$$\sum_{t=1}^{T} F(x^*) - 2F(x_t) \le F(x_{T+1}) + \sum_{t=1}^{T} \frac{3L}{2}\|x_{t+1} - x_t\|^2 + \sum_{t=1}^{T} \langle \nabla F(x_t), x^* - x_{t+1}\rangle.$$

For the last term, we further decompose it as

$$\langle \nabla F(x_t), x^* - x_{t+1}\rangle \le \langle g_t, x^* - x_t\rangle + \langle \nabla F(x_t) - g_t, x^* - x_t\rangle + \|\nabla F(x_t)\| \|x_t - x_{t+1}\|.$$

For the second term $\langle \nabla F(x_t) - g_t, x^* - x_t \rangle$, we can use the property of the one-point gradient estimator and upper bound it at the level of $O(\delta_t)$. For controlling $\|x_t - x_{t+1}\|$, we carefully bound the magnitude of the shifted gradient estimator $\tilde{g}_{t,i}$ and tune the learning $\eta_t$.

For the upper bound of the term $\langle g_t, x^* - x_t \rangle$, we first show that the update rule is equivalent to the mirror descent with a regularizer

$$\Phi(x) = -2 \sum_{i=1}^k \sqrt{x_{t,i}} + \mathbb{I}_{\mathcal{X}}(x), \quad \mathcal{X} = \{x \in [0,1]^k : \sum_{i=1}^k x_i \le m\}.$$

In fact, the update rules of mirror descent, i.e. $\hat{y}_{t+1} = \nabla\Phi(x_t) - \eta_t (g_t + \epsilon_t), x_{t+1} = \nabla\Phi^*(\hat{y}_{t+1})$, with our definition of $\Phi$, is equivalent to

$$x_{t+1,i} = \nabla\Phi^*\left(\frac{-1}{\sqrt{x_{t,i}}} - \eta_t(\tilde{g}_{t,i} + \epsilon_{t,i})\right).$$

To see this, by the definition of $\Phi^*(y) = \sup_{x \in \mathcal{X}} \langle x, y \rangle + 2 \sum_{i=1}^k \sqrt{x_i}$, the Lagrangian is

$$\mathcal{L}(x, \lambda) = \langle y, x \rangle + 2 \sum_{i=1}^k \sqrt{x_i} - \lambda\left(\sum_{i=1}^k x_i - m\right).$$

We take the derivative and solve it to obtain $x_i^* = \frac{1}{(y_i - \lambda)^2}$, while $\sum_{i=1}^k \frac{1}{(y_i - \lambda)^2} \le m$. Hence,

$$x_{t+1,i} = \frac{1}{\left(\frac{1}{\sqrt{x_{t,i}}} + \eta_t(\tilde{g}_{t,i} + \epsilon_{t,i})\right)^2 + \lambda\left(\frac{1}{\sqrt{x_{t,i}}} + \eta_t(\tilde{g}_{t,i} + \epsilon_{t,i})\right) + \lambda^2},$$

with $\lambda \ge 0$ and $\sum_{i=1}^k \frac{1}{\left(\frac{1}{\sqrt{x_{t,i}}} + \eta_t(\tilde{g}_{t,i} + \epsilon_{t,i})\right)^2 + \lambda\left(\frac{1}{\sqrt{x_{t,i}}} + \eta_t(\tilde{g}_{t,i} + \epsilon_{t,i})\right) + \lambda^2} \le m.$

We then show that with our shifted gradient estimator $\tilde{g}_{t,i}$, it suffices to take $\lambda = 0$ to ensure $x_{t+1,i}$ stays in the feasible range. Then, we use a result from Zimmert & Marinov (2024), which shows that there exist an appropriate $\epsilon_{t,i}$ under a careful choice of $\eta_t$ and the design of $\tilde{g}_{t,i}$, that ensures the equivalence to the update rule

$$x_{t+1,i} = \frac{1}{\left(\frac{1}{\sqrt{x_{t,i}}} + \eta_t(\tilde{g}_{t,i} + \epsilon_{t,i})\right)^2} = x_{t,i}\left(1 - 2\eta_t\sqrt{x_{t,i}}\tilde{g}_{t,i}\right).$$

Armed with the equivalence, we employ the classical result of regret for mirror descent (e.g. Theorem 5.6 of Hazan et al. (2016)) to bound it as

$$\sum_{t=1}^T \langle g_t, x^* - x_t \rangle \le \frac{D_\Phi(x^*, x_1)}{\eta_{t+1}} + \sum_{t=1}^T \eta_t O\left(\|g_t + \epsilon_t\|^2\right).$$

Combining the results and tuning the parameters needed, we obtain a regret for the continuous relaxation $F$. Lastly, we utilize the definition of the continuous relaxation $F$ and obtain the guarantees in the original discrete space stated in Theorem 5.1. □

# 6 STRATEGIC EXPERTS PROBLEM UNDER SEMI-BANDIT FEEDBACK

In this section, we show that the regret bound of our algorithm can be improved when we have access to the losses $\ell_{t,i}, \forall t \in [T]$ if $i \in S_t$. We also assume that the utility function takes the form of $f(S_t, r_t) = -\sum_{i \in S_t} \ell(p_{t,i}, r_t)$, where $\ell \in [-1, 1]$, in which case the learner can construct $f(S_t, r_t)$ based on the feedbacks received. Notice that now the utility function $f$ is no longer monotone and we allow the subset $S$ to contain more than $m$ experts.

The main ingredients of our algorithm remain the same. With the specific form of $f$, we then consider a continuous relaxation through the multilinear extension, which is

$$F(x, r) = \sum_{S \in \mathcal{S}} \prod_{i \in S} x_i \prod_{j \notin S} (1 - x_j) f(S, r).$$

We then sample the subset $S$ by including each expert $i$ with a probability of $x_i$.

With this, we can strengthen the result in Theorem 5.1 (which only upper bound the regret in expectation) to a high probability result. Specifically, we show that with a probability of at least $1 - \zeta$,

$$\sum_{t=1}^{T} f(S^*) - 2f(S_t) = \left(\sum_{t=1}^{T} \langle \ell_t, x_t \rangle - \min_x \sum_{t=1}^{T} \langle \ell_t, x \rangle \right) + O\left(\log(k/\zeta)\sqrt{T}\right) .$$

In this case, we can build the loss estimate for each expert individually, using importance sampling $g_{t,i} = \frac{\ell(p_{t,i}, r_t) \mathbb{I}\{i \in S_t\}}{x_{t,i} + \delta_t}$. By leveraging existing results from online learning, we balance between the cumulative bias and variance by balancing $O(\sum_{t=1}^{T} \delta_t)$ and $O(1/\delta_T)$. Overall, our algorithm is summarized in the following Algorithm.

---

**Algorithm 2:** Algorithm for Semi Bandit Feedback

**Input:** Learning rate $\eta_t$, parameter $\delta_t$

1 **for** $t = 1, \ldots, T$ **do**

2 $\quad$ Sample $S_t$ from $x_t$, where each expert $i$ is included in $S_t$ with probability $x_i$.

3 $\quad$ Receive $\ell(p_{t,i}, r_t), \forall i \in S_t$.

4 $\quad$ Set $g_{t,i} = \frac{\ell(p_{t,i}, r_t) \mathbb{I}\{i \in S_t\}}{x_{t,i} + \delta_t}$.

5 $\quad$ Set $\tilde{g}_{t,i} = \hat{g}_{t,i} + \frac{\mathbb{I}\{i \in S_t\}}{x_{t,i} + \delta_t}$.

6 $\quad$ Update $x_{t+1,i} = x_{t,i} \left(1 - 2\eta_t \sqrt{x_{t,i}} \tilde{g}_{t,i}\right)$.

---

**Theorem 6.1.** *With $\eta_t = \frac{1}{t^{3/4}}, \delta_t = \frac{4}{t^{1/4}}$, and a probability of at least $1 - \zeta$, we have*

$$\sum_{t=1}^{T} f(S^*) - 2f(S_t) \le O\left(D_\Phi(x^*, x_1) T^{3/4} + \left(m^2 + k + \ln(k/\zeta)\right) T^{3/4} + \sqrt{\ln(k/\zeta)T}\right) .$$

## 7 EXPERIMENTS

In this section, we validate our algorithm under full bandit feedback with synthetic data. We implement our algorithm with a learning rate of $0.005$, and gradient estimation parameter $\delta_t = 0.1$. We plot the averaged regret (note that this is not the $\alpha$-approximate regret) and the difference between expert predictions and their private belief in Figure 1. We evaluated our algorithm under the configurations of $k = 4, 6, 8$ and $m = 2, 3$. For each experiment, we set the time horizon to $10000$. We also implemented the classic online mirror descent algorithm with a learning rate of $0.01$. For each configuration, we perform the experiment with 5 different random seeds and the mean and standard deviation are reported in the figure. We ran all of the experiments on Google Colab with 12.7 GB System RAM.

For the binary realization $r_t$ and private belief $\{b_{t,i}\}_{t=1}^{T}, \forall i \in [k]$, we randomly generate them before the interaction of between the experts and the algorithm. To implement the strategic experts, we let each expert to randomly deviate from their private belief with a magnitude of $(b_{t-1,i} - r_t)$ with a probability of $1 - x_{t-1,i}$.

We can see that in all settings, our algorithm can effectively elicit truthful responses from the experts, as the differences between the expert predictions and the private belief go to 0 as the time step increases. Compared to the online mirror descent algorithm, our algorithm converges slower, but to a better approximation ratio. It also achieves higher average utility when compared to the online mirror descent. Moreover, it is clear that the online mirror descent is not incentive-compatible, as the differences between the expert predictions and their private beliefs are constant. We can also conclude that our algorithm can minimize the regret, and the empirical approximation ratio is much better than our theoretical value of $1/2$.

## 8 CONCLUSION

In this paper, we examine the combinatorial online prediction problem with strategic experts and bandit feedback. To maximize their chances of being selected, the experts may strategically manip-

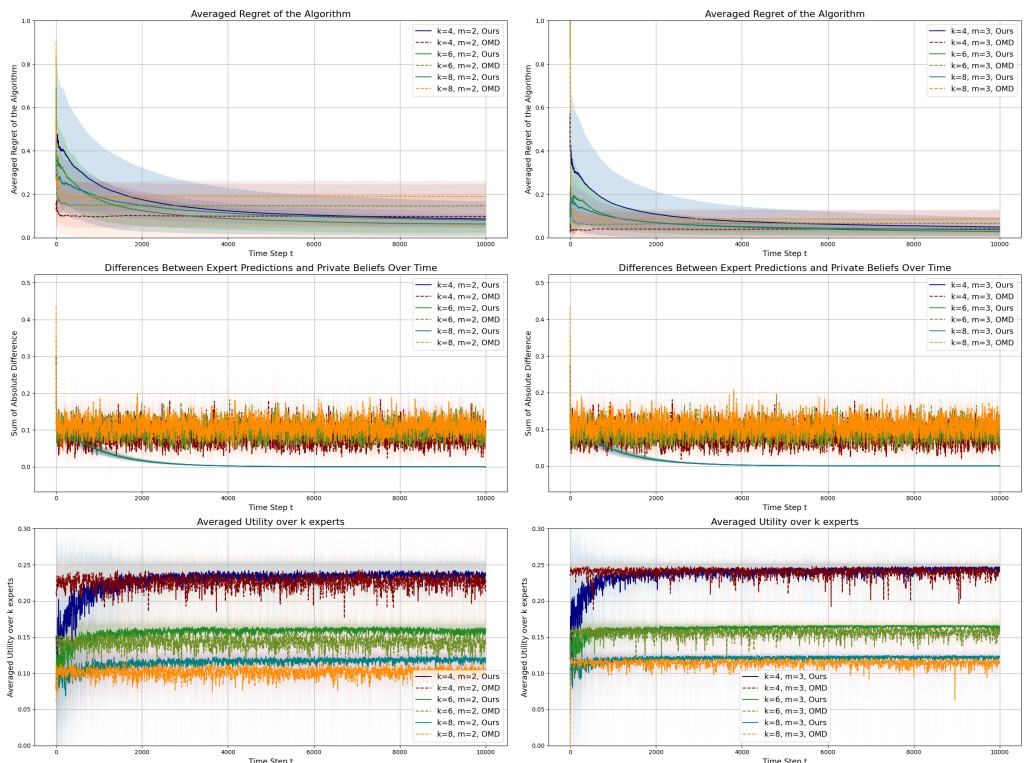

Figure 1: The averaged regret, the difference between expert predictions and private beliefs, and the averaged utility over time of our algorithm compared to the online mirror descent algorithm with $k = [4, 6, 8]$ and $m = [2, 3]$.

ulate their predictions about a series of binary events. Our learning algorithm achieves two objectives simultaneously. The first is to be no-regret, which is guaranteed with a regret of order $O(T^{(3/4)})$. The second is to ensure incentive compatibility, which ensures that each expert's best strategy is to accurately report their true beliefs about the outcomes of each event. Our algorithm therefore both provides high-accuracy and robust predictions and also promotes honesty from the experts which is beneficial to society. We validate the effectiveness of our algorithm through empirical evaluations with a synthetic dataset.

There are several future directions that might be interesting to the community. One direction is to extend the guarantees to any utility function, which is to lift or relax Assumption 4.1. Another possible future work is to develop an incentive-compatible algorithm when the experts are long-term planners, which may behave strategically to maximize the cumulative return for multiple steps. This will induce a wider set of strategic behaviors from experts, which the algorithm design will need to handle carefully.

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

## A  MOTIVATING EXAMPLES

**Narrative Generation with Generative Models**  In this scenario, there are $k$ generative models tasked with creating engaging narratives, such as short stories or scripts. At each round $t \in [T]$, the learner receives information about current genre trends and reader preferences, which influences narrative suggestions. Each model then generates a narrative proposal and assigns a quality score based on the likelihood of keeping readers interested. The learner algorithm can select up to $m$ models in each round to maximize the overall appeal of the final narrative. To maximize their chances of being selected and potential payment, the generative models may engage with strategic behaviors.

**Online paging problem with advice**  In an online paging problem, there is a library of $N$ files and a cache with limited storage that can hold $m$ files at any time. At each round $t \in [T]$, a user requests one file and the learner algorithm must select $m$ out of $k$ experts, each of whom can observe the user history to make a probabilistic prediction of the next file requested. The learner's prediction then depends on the best prediction out of the $m$ experts.

## B  PROOF FOR THE FULL-BANDIT FEEDBACK SETTING

**Theorem 5.1.** *With* $\eta_t = \frac{1}{kt^{3/4}}, \delta_t = \frac{4}{t^{1/4}}$ *and* $f$ *satisfying assumption 4.1, we have*

$$\mathbb{E}\left[\sum_{t=1}^{T} f(S^*) - 2f(S_t)\right] \leq O\left(D_\Phi(x^*, x_1)k^{3/2}T^{3/4} + m^2kT^{3/4} + mLT^{3/4} + kGT^{3/4}\right) .$$

*Proof.* By the sampling procedure, we have $\mathbb{E}[f(S_t)] = F(\tilde{x}_t)$. Therefore, by Theorem B.1, we have

$$\mathbb{E}\left[\sum_{t=1}^{T} f(S^*) - 2f(S_t)\right] \leq O\left(D_\Phi(x^*, x_1)kT^{2/3} + m^2kGT^{2/3} + mLT^{2/3}\right) .$$

$\square$

**Theorem B.1.** *With* $\eta_t = \frac{1}{k^{3/2}t^{3/4}}, \delta_t = \frac{4}{t^{1/4}}$, *we have*

$$\sum_{t=1}^{T} F(x^*) - 2F(\tilde{x}_t) \leq O\left(D_\Phi(x^*, x_1)k^{3/2}T^{3/4} + m^2kT^{3/4} + mLT^{3/4} + kGT^{3/4}\right) .$$

*Proof.* Using Lemma B.3, we have

$$\sum_{t=1}^{T} F(x^*) - 2F(x_t) \leq F(x_{T+1}) + \sum_{t=1}^{T} \frac{3L}{2}\|x_{t+1} - x_t\|^2 + \sum_{t=1}^{T} \langle \nabla F(x_t), x^* - x_{t+1} \rangle .$$

For the last term, we have

$$\langle \nabla F(x_t), x^* - x_{t+1} \rangle = \langle g_t, x^* - x_t \rangle + \langle \nabla F(x_t) - g_t, x^* - x_t \rangle + \langle \nabla F(x_t), x_t - x_{t+1} \rangle$$
$$\leq \langle g_t, x^* - x_t \rangle + \langle \nabla F(x_t) - g_t, x^* - x_t \rangle + \|\nabla F(x_t)\| \|x_t - x_{t+1}\| .$$

For the last term, if $(1 - \gamma_t)(m - 1) \leq \frac{4\eta_t k^2}{3\delta_t}$, we have

$$x_{t,i} - x_{t+1,i} = 2\eta_t x_{t,i}\sqrt{x_{t,i}}\tilde{g}_{t,i}$$
$$= 2\eta_t x_{t,i}\sqrt{x_{t,i}}\left(g_{t,i} + \frac{k}{2\delta_t}\right)$$
$$= 2\eta_t x_{t,i}\sqrt{x_{t,i}}g_{t,i} + \frac{\eta_t k x_{t,i}\sqrt{x_{t,i}}}{\delta_t}$$
$$\leq \left(\frac{2\eta_t k}{\delta_t}\right)x_{t,i} .$$

where the second inequality follows from $g_{t,i} \leq \frac{k}{2\delta_t}$. Hence $\|x_t - x_{t+1}\| \leq \frac{2\eta_t mk}{\delta_t}$. Therefore,

By Lemma B.1, we have

$$
\begin{aligned}
\mathbb{E}\left[\langle \nabla F(x_t) - g_t, x^* - x_t \rangle\right] &= \mathbb{E}\left[\langle \nabla F(x_t) - \nabla F(\tilde{x}_t), x^* - x_t \rangle\right] \\
&\leq m\mathbb{E}\left[\|\nabla F(x_t) - \nabla F(\tilde{x}_t)\|\right] \\
&\leq mL\mathbb{E}\left[\|x_t - \tilde{x}_t\|\right] \\
&\leq O\left(\delta_t mL\right).
\end{aligned}
$$

By Theorem 5.6 of Hazan et al. (2016), we have

$$
\sum^T \langle g_t, x^* - x_t \rangle \leq \frac{D_\Phi(x^*, x_1)}{\eta_{t+1}} + \sum_{t=1}^T \eta_t O\left(\|g_t + \epsilon_t\|^2\right).
$$

Using the definition of $\epsilon_t$, Lemma B.4, we have

$$
\|g_t + \epsilon_t\| \leq \|g_t\| + \|\epsilon_t\| \leq \|g_t\| + \|\eta_t \sqrt{x_t} \tilde{g}_t\|.
$$

For the second term, we have

$$
\begin{aligned}
\eta_t \sqrt{x_{t,i}} \tilde{g}_{t,i} &= \eta_t \sqrt{x_{t,i}} \left(g_{t,i} + \frac{k}{2\delta_t}\right) \\
&\leq \eta_t g_{t,i} + \frac{\eta_t k}{\delta_t} \\
&\leq \frac{\eta_t k}{2\delta_t},
\end{aligned}
$$

where the inequalities follows from $g_{t,i} \leq \frac{k}{2\eta_t}$.

Thus we have

$$
\begin{aligned}
&\sum_{t=1}^T \langle \nabla F(x_t), x^* - x_{t+1} \rangle \\
&= \sum_{t=1}^T \left(\langle g_t, x^* - x_t \rangle + \langle \nabla F(x_t) - g_t, x^* - x_t \rangle + \|\nabla F(x_t)\| \|x_t - x_{t+1}\|\right) \\
&\leq \frac{D_\Phi(x^*, x_1)}{\eta_{T+1}} + \sum_{t=1}^T O\left(\delta_t mL + \frac{\eta_t mk}{\delta_t} + \eta_t \|g_t + \epsilon_t\|^2\right) \\
&\leq \frac{D_\Phi(x^*, x_1)}{\eta_{T+1}} + \sum_{t=1}^T O\left(\delta_t mL + \frac{\eta_t mk}{\delta_t} + \frac{\eta_t k^3}{\delta_t^2}\right).
\end{aligned}
$$

Therefore,

$$
\begin{aligned}
&\sum_{t=1}^T F(x^*) - 2F(x_t) \\
&\leq F(x_{T+1}) + \frac{D_\Phi(x^*, x_1)}{\eta_{T+1}} + \sum_{t=1}^T O\left(\frac{\eta_t^2 m^2 k^2}{\delta_t^2} + \delta_t mL + \frac{\eta_t mk}{\delta_t} + \frac{\eta_t k^3}{\delta_t^2}\right).
\end{aligned}
$$

By our choice of $\eta_t, \delta_t$, we have

$$
\sum_{t=1}^T F(x^*) - 2F(x_t) \leq O\left(D_\Phi(x^*, x_1) k^{3/2} T^{3/4} + m^2 k T^{3/4} + mL T^{3/4}\right).
$$

Lastly,

$$\sum_{t=1}^{T} F(x^*) - 2F(\tilde{x}_t) = \sum_{t=1}^{T} F(x^*) - 2F(x_t) + O\left(kG\sum_{t=1}^{T}\delta_t\right)$$

$$= O\left(D_\Phi(x^*, x_1)k^{3/2}T^{3/4} + m^2 kT^{3/4} + mLT^{3/4} + kGT^{3/4}\right).$$

$\square$

**Lemma B.1.** *Define* $\tilde{F}(x, r) = \mathbb{E}_z\left[F(x + \delta_t z, r)\right]$. *Then*

$$\mathbb{E}_{\partial[-\delta_t, +\delta_t]^k}\left[F(x + \delta_t z, r)z\right] = \frac{2\delta_t}{k}\nabla\tilde{F}(x, r).$$

*Proof.* Notice that

$$\nabla\int_{[-\delta_t, +\delta_t]^k} F(x + \delta_t z, r)dv = \int_{\partial[-\delta_t, +\delta_t]^k} F(x + \delta_t z, r)n(z)dS,$$

where $\partial[-\delta_t, +\delta_t]^k$ denotes the surface of the hypercube, $n(z)$ is the outward normal vector at point $z$ on the surface and $dS$ is the surface element. Taking expectation over $z$, we have

$$\mathbb{E}_z\left[F(x + \delta_t z)\right] = \frac{\int_{[-\delta_t, +\delta_t]^k} F(x + \delta_t z, r)dv}{\text{Vol of hypercube}}.$$

For a hypercube of side length $2\delta_t$, the volume grow as $(2\delta_t)^k$ and the surface area grows as $2d(2_t^\delta)^{k-1}$. Using the ratio of volume to surface is $\frac{2\delta_t}{k}$, we have the result. $\square$

**Lemma B.2** (Vondrák et al. (2011), Lemma 3.2).

$$\langle x - y, \nabla F(x)\rangle \leq 2F(x) - F\left(\max\{x, y\}\right) - F\left(\min\{x, y\}\right)$$

**Lemma B.3.**

$$\sum_{t=1}^{T} F(x^*) - 2F(x_t) \leq F(x_{T+1}) + \sum_{t=1}^{T}\frac{3L}{2}\|x_{t+1} - x_t\|^2 + \sum_{t=1}^{T}\langle\nabla F(x_t), x^* - x_{t+1}\rangle.$$

*Proof.* By the smoothness of $F$, we have

$$F(x_t) \leq F(x_{t+1}) + \langle\nabla F(x_{t+1}), x_t - x_{t+1}\rangle + \frac{L}{2}\|x_{t+1} - x_t\|^2$$

$$= F(x_{t+1}) + \langle\nabla F(x_{t+1}) - \nabla F(x_t), x_t - x_{t+1}\rangle + \langle\nabla F(x_t), x_t - x_{t+1}\rangle + \frac{L}{2}\|x_{t+1} - x_t\|^2$$

$$= F(x_{t+1}) + \langle\nabla F(x_t), x_t - x_{t+1}\rangle + \|\nabla F(x_{t+1}) - \nabla F(x_t)\|\,\|x_t - x_{t+1}\| + \frac{L}{2}\|x_{t+1} - x_t\|^2$$

$$\leq F(x_{t+1}) + \langle\nabla F(x_t), x_t - x_{t+1}\rangle + \frac{3L}{2}\|x_{t+1} - x_t\|^2.$$

Rearranging the terms, we have

$$F(x_{t+1}) \geq F(x_t) + \langle\nabla F(x_t), x^* - x_t\rangle + \langle\nabla F(x_t), x_{t+1} - x^*\rangle - \frac{3L}{2}\|x_{t+1} - x_t\|^2,$$

and

$$\langle\nabla F(x_t), x_t - x^*\rangle + F(x_{t+1}) - F(x_t) \geq \langle\nabla F(x_t), x_{t+1} - x^*\rangle - \frac{3L}{2}\|x_{t+1} - x_t\|^2.$$

By Lemma B.2, we have

$$\langle\nabla F(x_t), x_t - x^*\rangle + F(x_{t+1}) - F(x_t) \leq F(x_t) - F\left(\max\{x_t, x^*\}\right) - F\left(\min\{x_t, x^*\}\right) + F(x_{t+1})$$

$$\leq F(x_t) + F(x_{t+1}) - F\left(\max\{x_t, x^*\}\right)$$

$$\leq F(x_t) + F(x_{t+1}) - F(x^*),$$

Hence, we have

$$F(x^*) - F(x_t) - F(x_{t+1}) \leq \langle \nabla F(x_t), x_{t+1} - x^* \rangle - \frac{3L}{2} \|x_{t+1} - x_t\|^2 .$$

Therefore,

$$\sum_{t=1}^{T} F(x^*) - 2F(x_t) = F(x_{T+1}) - F(x_1) + \sum_{t=1}^{T} (1 - \|x_t\|_\infty) F(x^*) - F(x_t) - F(x_{t+1})$$

$$\leq F(x_{T+1}) + \sum_{t=1}^{T} \frac{3L}{2} \|x_{t+1} - x_t\|^2 + \sum_{t=1}^{T} \langle \nabla F(x_t), x^* - x_{t+1} \rangle .$$

$\square$

**Lemma B.4** (Lemma 7 of Zimmert & Marinov (2024)). *If $|\eta_t \sqrt{x_{t,i}} \tilde{g}_{t,i}| \leq \frac{1}{4}, \forall t \in [T], i \in [K]$, then there is an $\epsilon_{t,i} = O\left(\eta_t \sqrt{x_{t,i}} \tilde{g}_{t,i}\right)$ such that*

$$x_{t+1,i} = \frac{1}{\left(\frac{1}{\sqrt{x_{t,i}}} + \eta_t \left(\tilde{g}_{t,i} + \epsilon_{t,i}\right)\right)^2} = x_{t,i} \left(1 - 2\eta_t \sqrt{x_{t,i}} \tilde{g}_{t,i}\right) .$$

**Lemma B.5.** *Suppose $\frac{\eta_t k}{\delta_t} + (1 - \gamma_t)(k - 1) \leq 1$. Define $\Phi$ to be the potential function plus the indicator function of the hypercube $[0,1]^k$, i.e. $\Phi(x) = -2 \sum_{i=1}^{k} \sqrt{x_{t,i}} + \mathbb{I}_\mathcal{X}(x)$, $\mathcal{X} = \{x \in [0,1]^k : \sum_{i=1}^{k} x_i \leq m\}$. Then the update rule $x_{t+1,i} = x_{t,i} \left(1 - 2\eta_t \sqrt{x_{t,i}} \tilde{g}_{t,i}\right)$ is equivalent to the following update*

$$\hat{y}_{t+1} = \nabla \Phi(x_t) - \eta_t \left(g_t + \epsilon_t\right) ,$$
$$x_{t+1} = \nabla \Phi^* \left(\hat{y}_{t+1}\right) .$$

*Proof.* Using the definition of $\Phi$, we have

$$-\hat{y}_{t+1,i} = \frac{1}{\sqrt{x_{t,i}}} + \eta_t \left(g_{t,i} + \epsilon_{t,i}\right) .$$

By the definition of $\Phi^*(y) = \sup_{x \in \mathcal{X}} \langle x, y \rangle + 2 \sum_{i=1}^{k} \sqrt{x_i}$, the Lagrangian is $\mathcal{L}(x, \lambda) = \langle y, x \rangle + 2 \sum_{i=1}^{k} \sqrt{x_i} - \lambda \left(\sum_{i=1}^{k} x_i - m\right)$. We can take the derivative and solve it to obtain $x_i^* = \frac{1}{(y_i - \lambda)^2}$, while $\sum_{i=1}^{k} \frac{1}{(y_i - \lambda)^2} \leq m$. As $\nabla \Phi^*$ is invariant under constant vector perturbation, we therefore have

$$x_{t+1,i} = \nabla \Phi^* \left(\frac{-1}{\sqrt{x_{t,i}}} - \eta_t \left(g_{t,i} + \epsilon_{t,i}\right)\right)$$

$$= \nabla \Phi^* \left(\frac{-1}{\sqrt{x_{t,i}}} - \eta_t \left(\tilde{g}_{t,i} + \epsilon_{t,i}\right)\right)$$

$$= \frac{1}{\left(\frac{1}{\sqrt{x_{t,i}}} + \eta_t \left(\tilde{g}_{t,i} + \epsilon_{t,i}\right)\right)^2 + \lambda \left(\frac{1}{\sqrt{x_{t,i}}} + \eta_t \left(\tilde{g}_{t,i} + \epsilon_{t,i}\right)\right) + \lambda^2} ,$$

with $\lambda$ satisfy $\lambda \geq 0$ and $\sum_{i=1}^{k} \frac{1}{\left(\frac{1}{\sqrt{x_{t,i}}} + \eta_t (\tilde{g}_{t,i} + \epsilon_{t,i})\right)^2 + \lambda \left(\frac{1}{\sqrt{x_{t,i}}} + \eta_t (\tilde{g}_{t,i} + \epsilon_{t,i})\right) + \lambda^2} \leq m.$

We next want to show that $\lambda = 0$, and we want to apply Lemma B.4,

$$x_{t+1,i} = \frac{1}{\left(\frac{1}{\sqrt{x_{t,i}}} + \eta_t \left(\tilde{g}_{t,i} + \epsilon_{t,i}\right)\right)^2} = x_{t,i} \left(1 - 2\eta_t \sqrt{x_{t,i}} \tilde{g}_{t,i}\right) .$$

We first show that $\sum_{i=1}^{k} x_{t+1,i} \leq m$.

We prove this by induction, suppose $x_{t,i} \in [0,1]$ and $\sum_{i=1}^{k} x_{t,i} \leq m$, then

$$
\begin{aligned}
\sum_{i=1}^{k} x_{t+1,i} &= \sum_{i=1}^{k} x_{t,i} - 2\eta_t \sum_{i=1}^{k} x_{t,i}\sqrt{x_{t,i}}\tilde{g}_{t,i} \\
&= \sum_{i=1}^{k} x_{t,i} - 2\eta_t \sum_{i=1}^{k} x_{t,i}\sqrt{x_{t,i}}g_{t,i} - 2\eta_t \sum_{i=1}^{k} x_{t,i}\sqrt{x_{t,i}}\left(\frac{k}{2\delta_t}\right) \\
&= \sum_{i=1}^{k} x_{t,i} + 2\eta_t \sum_{i=1}^{k} x_{t,i}\sqrt{x_{t,i}}\left(\frac{kf(S_t,r_t)z_{t,i}}{2\delta_t}\right) - \frac{\eta_t k}{\delta_t}\sum_{i=1}^{k} x_{t,i}\sqrt{x_{t,i}} \\
&\leq \sum_{i=1}^{k} x_{t,i} \\
&\leq m\,,
\end{aligned}
$$

where the inequality used the fact $\frac{kf(S_t,r_t)z_t}{2\delta_t} \leq \frac{\eta_t k}{\delta_t}$.

If $\frac{\eta_t k}{\delta_t} \leq 1/4$, then by Lemma B.6, we have $x_{t+1,i} \in [0,1]$. Further, we have $|\eta_t \sqrt{x_{t,i}}\tilde{g}_{t,i}| \leq \frac{1}{4}, \forall t \in [T], i \in [K]$.

Hence as long as $x_1$ are initialized appropriately, we have $\lambda = 0$ for each $t$.

$\square$

**Lemma B.6.** *Suppose* $\frac{\eta_t k}{\delta_t} \leq \frac{1}{4}$, *then* $|\eta_t \sqrt{x_{t,i}}\tilde{g}_{t,i}| \leq \frac{1}{4}, \forall t \in [T], i \in [K]$ *and* $x_{t,i} \in [0,1]$, $\forall t \in [T], i \in [K]$.

*Proof.* We assume $x_{t,i} \in [0,1]$ and show that $x_{t+1,i} \in [0,1]$. Then, with the proper initialization, we can have $x_{t,i} \in [0,1], \forall t \in [T]$ by induction. By the definition of $\tilde{g}_{t,i}, g_{t,i}$, we have

$$
\begin{aligned}
\eta_t \sqrt{x_{t,i}}\tilde{g}_{t,i} &= \eta_t \sqrt{x_{t,i}}\left(g_{t,i} + \frac{k}{2\delta_t}\right) \\
&\leq \frac{\eta_t k}{\delta_t} \\
&\leq 1/4\,,
\end{aligned}
$$

where the second to the last inequality holds as $g_{t,i} \leq \frac{k}{2\delta_t}$.

By the definition of $\tilde{g}_{t,i}, g_{t,i}$, we also have

$$
\begin{aligned}
-\eta_t \sqrt{x_{t,i}}\tilde{g}_{t,i} &= \eta_t \sqrt{x_{t,i}}\left(-g_{t,i} - \frac{k}{2\delta_t}\right) \\
&= \eta_t \sqrt{x_{t,i}}\left(\frac{k(S_t,r_t)z_{t,i}}{2\delta_t}\right) - \frac{\eta_t k \sqrt{x_{t,i}}}{2\delta_t} \\
&\leq 0\,,
\end{aligned}
$$

where the inequality follows from $\frac{k(S_t,r_t)z_{t,i}}{2\delta_t} \leq \frac{k}{2\delta_t}$.

Then by the definition of the update rule, we have $x_{t+1,i} \in [0,1]$. $\square$

**Theorem 5.2.** *Algorithm 1 is incentive-compatible.*

*Proof.* For expert $i$, if it wants to increase its chance of being selected at time $t+1$, it has to maximize $\tilde{x}_{t+1,i}$, which is the same as maximizing $x_{t+1,i} = x_{t,i}\left(1 - 2\eta_t \sqrt{x_{t,i}}\tilde{g}_{t,i}\right)$. Since $\tilde{g}_{t,i}$ is linear in $g_{t,i}$. Expert $i$ will then need to minimize $g_{t,i}$. By the definition $g_{t,i}$, it is equivalent to maximizing $F(\tilde{x}_t)$ in expectation. Using the definition of $F(\tilde{x}_t)$, it is equivalent to maximize $f(S_t)$ in expectation. As $f(S_t)$ is a affine function of $\ell(p_{t,i},r)$, expert $i$ needs to minimize $\ell(p_{t,i})$. If $\ell$ is a proper loss function, then it is in expert $i$'s best interest to submit $b_{t,i}$.

$\square$

## C  PROOF FOR THE SEMI-BANDIT FEEDBACK SETTING

**Lemma C.1.** *With probability of at least $1 - \zeta$, we have*

$$\sum_{t=1}^{T} \sum_{i \in S_t} \ell_{t,i} - \min_{S} \sum_{t=1}^{T} \sum_{i \in S} \ell_{t,i} = \left( \sum_{t=1}^{T} \langle \ell_t, x_t \rangle - \min_{x} \sum_{t=1}^{T} \langle \ell_t, x \rangle \right) + O\left(\log(k/\zeta)\sqrt{T}\right) .$$

*Proof.* Let $\ell_t$ be a vector where the $i$-th entry is $\ell_{t,i}$. Since the function is linear we have

$$\min_{x} \sum_{t=1}^{T} \langle \ell_t, x \rangle = \min_{S} \sum_{t=1}^{T} \sum_{i \in S} \ell_{t,i} .$$

Then, using Martingale concentration, with a probability of at least $1 - \delta$,

$$\sum_{t=1}^{T} \langle \ell_t, x_t \rangle - \sum_{t=1}^{T} \sum_{i \in S_t} \ell_{t,i} = \sum_{t=1}^{T} \sum_{i \in [k]} \left( \mathbb{E}_t \left[ \mathbb{I}[i \in S_t] \right] - \mathbb{I}[i \in S_t] \right)$$

$$= \sum_{i \in [k]} \sum_{t=1}^{T} \left( \mathbb{E}_t \left[ \ell_{t,i} \mathbb{I}[i \in S_t] \right] - \ell_{t,i} \mathbb{I}[i \in S_t] \right)$$

$$\leq O\left(\log(k/\zeta)\sqrt{T}\right) .$$

$\square$

**Lemma C.2.** *Define $\Phi$ to be the potential function plus the indicator function of the hypercube $[0,1]^k$, i.e. $\Phi(x) = -2\sum_{i=1}^{k}\sqrt{x_{t,i}} + \mathbb{I}_{\mathcal{X}}(x)$, $\mathcal{X} = \{x \in [0,1]^k : \sum_{i=1}^{k} x_i \leq m\}$. Then the update rule $x_{t,i}\left(1 - 2\eta_t\sqrt{x_{t,i}}\tilde{g}_{t,i}\right)$ is equivalent to the following update*

$$\hat{y}_{t+1} = \nabla\Phi(x_t) - \eta_t\left(g_t + \epsilon_t\right) ,$$
$$x_{t+1} = \nabla\Phi^*\left(\hat{y}_{t+1}\right) ,$$

*with $\gamma_t = \frac{\eta_{t+1}}{\eta_t}$*

*Proof.* Using the definition of $\Phi$, we have

$$-\hat{y}_{t+1,i} = \frac{1}{\sqrt{x_{t,i}}} + \eta_t\left(g_{t,i} + \epsilon_{t,i}\right) .$$

By the definition of $\Phi^*(y) = \sup_{x \in \mathcal{X}} \langle x, y \rangle + 2\sum_{i=1}^{k}\sqrt{x_i}$, the Lagrangian is $\mathcal{L}(x, \lambda) = \langle y, x \rangle + 2\sum_{i=1}^{k}\sqrt{x_i} - \lambda\left(\sum_{i=1}^{k} x_i - m\right)$. We can take the derivative and solve it to obtain $x_i^* = \frac{1}{(y_i - \lambda)^2}$, while $\sum_{i=1}^{k} \frac{1}{(y_i - \lambda)^2} \leq m$. As $\nabla\Phi^*$ is invariant under constant vector perturbation, we therefore have

$$x_{t+1,i} = \nabla\Phi^*\left(\frac{-1}{\sqrt{x_{t,i}}} - \eta_t\left(g_{t,i} + \epsilon_{t,i}\right)\right)$$

$$= \nabla\Phi^*\left(\frac{-1}{\sqrt{x_{t,i}}} - \eta_t\left(\tilde{g}_{t,i} + \epsilon_{t,i}\right)\right)$$

$$= \frac{1}{\left(\frac{1}{\sqrt{x_{t,i}}} + \eta_t\left(\tilde{g}_{t,i} + \epsilon_{t,i}\right)\right)^2 + \lambda\left(\frac{1}{\sqrt{x_{t,i}}} + \eta_t\left(\tilde{g}_{t,i} + \epsilon_{t,i}\right)\right) + \lambda^2} ,$$

with $\lambda$ satisfy $\lambda \geq 0$ and $\sum_{i=1}^{k} \frac{1}{\left(\frac{1}{\sqrt{x_{t,i}}} + \eta_t(\tilde{g}_{t,i} + \epsilon_{t,i})\right)^2 + \lambda\left(\frac{1}{\sqrt{x_{t,i}}} + \eta_t(\tilde{g}_{t,i} + \epsilon_{t,i})\right) + \lambda^2} \leq m.$

We next want to show that $\lambda = 0$, and we want to apply Lemma B.4,

$$x_{t+1,i} = \frac{1}{\left(\frac{1}{\sqrt{x_{t,i}}} + \eta_t \left(\tilde{g}_{t,i} + \epsilon_{t,i}\right)\right)^2} = x_{t,i}\left(1 - 2\eta_t\sqrt{x_{t,i}}\tilde{g}_{t,i}\right).$$

We prove this by induction, suppose $x_{t,i} \in [0,1]$ and $\sum_{i=1}^k x_{t,i} \leq m$, then

$$\begin{aligned}
\sum_{i=1}^k x_{t+1,i} &= \sum_{i=1}^k x_{t,i} - 2\eta_t \sum_{i=1}^k x_{t,i}\sqrt{x_{t,i}}\tilde{g}_{t,i} \\
&= \sum_{i=1}^k x_{t,i} - 2\eta_t \sum_{i=1}^k x_{t,i}\sqrt{x_{t,i}}\left(\frac{\ell(p_{t,i},r_t)\mathbb{I}\{i \in S_t\}}{x_{t,i}+\delta_t} - \frac{\mathbb{I}\{i \in S_t\}}{x_{t,i}+\delta_t}\right) \\
&\leq \sum_{i=1}^k x_{t,i} \\
&\leq m.
\end{aligned}$$

If $\frac{\eta_t}{\delta_t} \leq 1/4$, then by Lemma C.3, we have $x_{t+1,i} \in [0,1]$. Further, we have $|\eta_t\sqrt{x_{t,i}}\tilde{g}_{t,i}| \leq \frac{1}{4}, \forall t \in [T], i \in [K]$.

Hence as long as $x_1$ are initialized appropriately, we have $\lambda = 0$ for each $t$. $\qquad \square$

**Lemma C.3.** *Suppose* $|\frac{\eta_t}{\delta_t}| \leq 1/4$*, then* $|\eta_t\sqrt{x_{t,i}}\tilde{g}_{t,i}| \leq \frac{1}{4}, \forall t \in [T], i \in [K]$ *and* $x_{t,i} \in [0,1]$*,* $\forall t \in [T], i \in [K]$*.*

*Proof.* We assume $x_{t,i} \in [0,1]$ and show that $x_{t+1,i} \in [0,1]$. Then, with the proper initialization, we can have $x_{t,i} \in [0,1], \forall t \in [T]$ by induction. By the definition of $\tilde{g}_{t,i}, g_{t,i}$, and the choice of $\eta_t, \delta_t$, we have

$$\begin{aligned}
\eta_t\sqrt{x_{t,i}}\tilde{g}_{t,i} &= \eta_t\sqrt{x_{t,i}}\left(\frac{\ell(p_{t,i},r_t)\mathbb{I}\{i \in S_t\}}{x_{t,i}+\delta_t} - \frac{\mathbb{I}\{i \in S_t\}}{x_{t,i}+\delta_t}\right) \\
&\leq \frac{\eta_t\sqrt{x_{t,i}}}{x_{t,i}+\delta_t} \\
&\leq \frac{\eta_t}{\delta_t} \\
&\leq 1/4.
\end{aligned}$$

By the definition of $\tilde{g}_{t,i}, g_{t,i}$, we also have

$$\begin{aligned}
\eta_t\sqrt{x_{t,i}}\tilde{g}_{t,i} &= \eta_t\sqrt{x_{t,i}}\left(\frac{\ell(p_{t,i},r_t)\mathbb{I}\{i \in S_t\}}{x_{t,i}-\delta_t} + \frac{\mathbb{I}\{i \in S_t\}}{x_{t,i}+\delta_t}\right) \\
&\geq \frac{-\eta_t}{\delta_t} \\
&\geq \frac{1}{4},
\end{aligned}$$

as each $\ell(p_{t,i},r_t) \in [0,1]$.

Then by the definition of the update rule, we have $x_{t+1,i} \in [0,1]$. $\qquad \square$

**Theorem 6.1.** *With* $\eta_t = \frac{1}{t^{3/4}}, \delta_t = \frac{4}{t^{1/4}}$*, and a probability of at least* $1 - \zeta$*, we have*

$$\sum_{t=1}^T f(S^*) - 2f(S_t) \leq O\left(D_\Phi(x^*, x_1)T^{3/4} + \left(m^2 + k + \ln(k/\zeta)\right)T^{3/4} + \sqrt{\ln(k/\zeta)T}\right).$$

*Proof.* By Theorem 5.6 of Hazan et al. (2016), we have

$$\sum^{T}\langle g_t, x^* - x_t\rangle \leq \frac{D_\Phi(x^*, x_1)}{\eta_{t+1}} + \sum_{t=1}^{T} \eta_t O\left(\|g_t + \epsilon_t\|^2\right).$$

By Lemma B.4, we have

$$
\begin{aligned}
g_{t,i} + \epsilon_{t,i} &\leq g_{t,i} + \eta_t \tilde{g}_{t,i} \\
&= g_{t,i} + \eta_t g_{t,i} + \left(\eta_t \frac{\mathbb{I}\{i \in S_t\}}{x_{t,i} + \delta_t}\right) \\
&\leq \frac{\mathbb{I}\{i \in S_t\}}{x_{t,i} + \delta_t} \\
&\leq \frac{\mathbb{I}\{i \in S_t\}}{\delta_t}.
\end{aligned}
$$

Summing over all $i$, we have $\|g_t + \epsilon_t\|^2 = \frac{m^2}{\delta_t^2}$.

Denote $x^* = \arg\min_x \sum_{t=1}^{T}\langle \ell_t, x\rangle$, by Lemma C.4 and Lemma C.5, with probability of at least $1 - \zeta$, we have

$$
\begin{aligned}
\sum_{t=1}^{T}\langle \ell_t, x^* - x_t\rangle &= \sum_{t=1}^{T}\langle g_t, x_t - x^*\rangle + \sum_{t=1}^{T}\langle \ell_t - g_t, x_t - x^*\rangle \\
&\leq \frac{D_\Phi(x^*, x_1)}{\eta_{t+1}} + \sum_{t=1}^{T} \frac{\eta_t m^2}{\delta_t^2} + O\left(\frac{\ln(k/\zeta)}{\delta_T} + k\sum_{t=1}^{T}\delta_t + \sqrt{\ln(k/\zeta)T}\right).
\end{aligned}
$$

Take $\eta_t = \frac{1}{t^{3/4}}$, $\delta_t = \frac{4}{t^{1/4}}$. Then, with a probability of at least $1 - \zeta$, we have

$$\sum_{t=1}^{T}\langle \ell_t, x_t - x^*\rangle \leq O\left(D_\Phi(x^*, x_1)T^{3/4} + \left(m^2 + k + \ln(k/\zeta)\right)T^{3/4} + \sqrt{\ln(k/\zeta)T}\right).$$

Then, using the definition of $f(S_t, r_t)$, we obtain the desired result

$\square$

**Lemma C.4** (Lemma 20 of Bai et al. (2020)). *Let $c_1, c_2, \ldots, c_t$ be fixed positive numbers. Then with probability at least $1 - \zeta$,*

$$\sum_{i=1}^{t} c_i \langle x_i, g_i - \ell_i\rangle = \mathcal{O}\left(k\sum_{i=1}^{t}\delta_i c_i + \sqrt{\ln(k/\zeta)\sum_{i=1}^{t}c_i^2}\right)$$

**Lemma C.5** (Lemma 21 of Bai et al. (2020)). *Let $c_1, c_2, \ldots, c_t$ be fixed positive numbers. Then for any $x^*$, with probability at least $1 - \zeta$,*

$$\sum_{i=1}^{t} c_i \langle x_i^\star, \ell_i - g_i\rangle = \mathcal{O}\left(\max_{i \leq t}\frac{c_i \ln(k/\zeta)}{\delta_t}\right)$$

