# OpenReview forum: "No-Regret and Incentive-Compatible Combinatorial Online Prediction"
_ICLR.cc/2025/Conference — Submitted to ICLR 2025_

### Official Review · Reviewer_v1cU · 2024-10-27

**Soundness:** 2
**Presentation:** 2
**Contribution:** 2
**Rating:** 3
**Confidence:** 3

**Summary:**

This paper considers incentive-compatible online learning with (semi-) bandit feedback. This problem extends the previous works by [Freeman et al., 2020, Zimmert and Marinov 2024] to the m-set reward setup. Specifically, in this work, the authors assume that at each round $t$, the learner can pick a subset with size at most $m$ and observe its reward defined by a submodular and monotone function. When the learner can only observe the reward of the whole chosen set, the authors propose and algorithm achieving O(T^{3/4}) 1/2-regret for the learner and ensuring incentive compatibility for each expert given the fact that their loss functions are proper. The algorithm is based on a combination of the BCO algorithm [Flaxman et al., 2005] and prod-like algorithm with Tsallis-1/2 regularizer [Zimmert and Marinov 2024]. The authors also consider the semi-bandit feedback setup with linear reward function but the guarantee seems to be the same. The authors also conduct experiments to show good empirical performance of their proposed algorithms.

**Strengths:**

- The problem considered in this paper is interesting and well-motivated.

- While the algorithm is based on a combination of the BCO algorithm [Flaxman et al., 2005] and prod-like algorithm with Tsallis-1/2 regularizer [Zimmert and Marinov 2024], this combination looks interesting to me. However, there are several places I do not understand and will clarify in the later section.

- The proof looks correct to me in general (with some points I do not understand to be mentioned later).

- Empirical results of their proposed algorithms are also shown to showcase the real performance.

**Weaknesses:**

- The algorithm description is not very clear to me. Several points are as follows:
  - What does $p$ mean in the notation $\mathbb{H}_r(p)$? What is the choice of $p$ in the algorithm?
  - What is the exact sampling scheme for $S_t$? While in Assumption 4.1, the authors claim certain properties for EXT, I do not understand the sampling scheme here. In addition, more examples of such EXT should be included to show that Algorithm 1 is effectively runnable.
  - No code is provided in the supplementary material.

- From the theoretical result side, one main disadvantage is the bad 1/2-approximation ratio of the regret, since 1-1/e is typical for monotone submodular maximization problem. Moreover, even in the linear reward case, the obtained results in Section 6 are 1/2-approximated regret, which is not ideal.

- As for the regret rate, while $T^{3/4}$ is the same as the BCO algorithm using one-point gradient estimation from [Flaxman et al., 2005],
  - there are more advanced algorithms for BCO achieving \sqrt{T} type regret (e.g. [Fokkema et al.,2024, Online Newton Method for Bandit Convex Optimisation]). I wonder whether these algorithms are applicable to this case?
  - for the linear case, the T^{3/4} rate may not be ideal. Also, I wonder why the first term in the bound in Theorem 6.1 is T^{2/3}? From the analysis, it seems to be still T^{3/4}.

- There are some writing typos:
  - Line 400: unfinished sentence.
  - Algorithm 2 title: not full-bandit feedback.
  - Line 477: T^{(3/4)} should be $O(T^{3/4}).

**Questions:**

Questions are listed in the "Weakness" section.

**Details Of Ethics Concerns:**

None.

---

### Official Review · Reviewer_5k8V · 2024-11-01

**Soundness:** 3
**Presentation:** 2
**Contribution:** 2
**Rating:** 5
**Confidence:** 2

**Summary:**

This work studies combinatorial prediction with bandit feedback, and the experts are strategic (which means that they do not necessarily report their true prediction). Therefore, the online learning algorithm to design has two objectives: 1) achieving low (approximate) regret and 2) incentivizing the experts to report ground truth.

In the one-point full-bandit setting, the proposed algorithm is incentive-compatible and ensures $O(T^{3/4})$ (2-approx-)regret. To deal with full-bandit feedback, one-point gradient estimation is performed on the continuous relaxation.

In the semi-bandit setting, a similar algorithm design enjoys improved regret bound, but the dominating term is still $O(T^{3/4})$.

**Strengths:**

This is the first set of incentive-compatible results in the combinatorial setting.

**Weaknesses:**

While this work definitely makes progress towards the combinatorial setting, their seems to be some assumptions needed. Firstly, I'm not sure about how strong they are. Moreover, since they are relatively heavy in maths, it would be great to give more intuition to help readers better understand them.

**Questions:**

1. It's a little wierd to have the bregman divergence term in the regret upper bound (Thm. 5.1, 6.1). Can't we just upper bound them?
2. In combinatorial MAB without incentive compatibility consideration, the regret can be $O(T^{1/2})$ [1]. Could the authors please explain why here we have to pay $O(T^{3/4})$?

[1] Audibert, J.-Y., Bubeck, S., and Lugosi, G. (2014). Regret in online combinatorial optimization. Mathematics of Operations Research, 39(1):31–45.

---

### Official Review · Reviewer_5j1e · 2024-11-01

**Soundness:** 3
**Presentation:** 2
**Contribution:** 2
**Rating:** 5
**Confidence:** 3

**Summary:**

This paper studies the problem of online combinatorial prediction problem with input from strategic experts, where the learner not only tries to maximize a submodular utility function under bandit or semi-bandit feedback, but also try to make the experts truthfully reporting their true predictions in the sense that being honest maximizes the probability of being chosen.

1. When only bandit feedback on the overall utility is available, the algorithm ensures an $1/2$-regret of $\mathcal O(T^{3/4})$ via an Online Mirror Descent with $1/2$-Tsallis entropy regularizer using the standard one-point gradient estimator.
2. When semi-bandit feedback on the losses of those chosen experts are available, the $1/2$-regret slightly improved -- although the dependency on $T$ is still $\mathcal O(T^{3/4})$, some of the terms replaced $T^{3/4}$ with $T^{2/3}$ or $T^{1/2}$.

**Strengths:**

1. The problem of incentive-compatible online learning is of great interest in the online learning literature.
2. The extension from standard online learning to combinatorial ones, and that from full information model to bandit feedback model, both look non-trivial.
3. The results are supplemented with numerical experiments.

**Weaknesses:**

1. [Results] The approximation factor is only $1/2$, which is usually far from the offline optimal one $1-c_f/e$. Even using the metric of $1/2$-regret, the algorithms are still only able to secure a $\mathcal O(T^{3/4})$ bound, which is much worse than the previous $\mathcal O(\sqrt T)$ ones.
2. [Techniques] Technically, the main innovation seems to be replacing the ordinary gradient descent (available in full-information setups, but can make the new action infeasible due to importance sampling under bandit feedback) with an OMD with $1/2$-Tsallis entropy regularizer. From the main text, it is hard to understand why it is hard to develop this technique here.
3. [Writing] The writing is often unclear. For example, the setup part is written in a pretty confusing way -- the word "each expert incurs a loss" sounds like the loss is the metric of experts instead of the learner; but it actually turns out that it is used to construct the submodular utility f(S,r) and experts are only interested in maximizing the probability of being selected. Also, what's the intuitive explanation of proper losses? Does it imply things more than "if your belief is correct, then being honest minimizes the loss"? I feel it'd be good to add some informal statements around such definitions.
4. [Related Work] The comparison with the literature is not well-organized and makes it hard to evaluate the role of this paper in the literature. For example, the comparison between this work and the closest one (Sadeghi & Fazel, 2024) was divided into Lines 104 -- 110 and Lines 150 -- 156, where the first part only mentioned loss functions & feedback models and the second one only says approximation factors & incentive-compatibility. I suggest the authors make a table to summarize the many previous works together with this paper, including setups (standard online learning or combinatorial one), feedback models, approximation factors, and when they're incentive-compatible, for a fair comparison with previous works.

Therefore, because of these issues, I do not feel it is possible to fully evaluate the real novelty of the contributions of the current version of this paper, hence my initial rating.

**Questions:**

See Weaknesses

---

### Official Review · Reviewer_g3cV · 2024-11-03

**Soundness:** 3
**Presentation:** 3
**Contribution:** 3
**Rating:** 6
**Confidence:** 3

**Summary:**

This paper studies an online bandit problem with strategic experts, where at each time the agent can use information up to m experts instead of 1 as in conventional adversarial bandit problems. Additionally, the experts are strategic, i.e., they may not report their true beliefs and will maximize the chance of being selected. The authors propose an algorithm that is both non-regret and incentive-compatible, which forces the experts to report truthfully without sacrificing in the long run. The proposed algorithm is validated on a synthetic dataset.

**Strengths:**

The paper is overall well-organized and sections are connected well. The literature review is clear and the preliminaries are concise. The goals, definitions, and settings are clear.

Section 4 provides ways of solving the problem layer by layer, from continuous approximations to one-point zenith order estimates. As a result, Algorithm 1 is naturally introduced.

Regret analyses under two different feedback settings are provided in Sections 5 and 6, respectively. The order looks correct and reasonable.

----

**Weaknesses:**

*1.* Assumption 4.1 looks like a very strong assumption. The extension mapping assumption might be hard to verify in practice.

*2.* The safety hypercube is used to ensure that perturbation points are always feasible. However, I don't see how p and r are used in the simulation section or can be estimated in practical applications.


*3.* The problem setting is not new; it extends a previous m-combinatorial bandit paper that used quadratic losses [Sadeghi & Fazel (2024)]. However, no simulation comparisons are provided for quadratic losses.


----

**Questions:**

*1.* Assumption 4.1 seems quite abstract. Can you explain why this assumption is necessary, i.e., is it due to your zeroth-order method? In other words, if there exists a different method, can the assumption be avoided?

*2.* How is the safety hypercube selected in practice? The gradient bias does not depend on m. Will the side length r implicit depend on m, i.e., the bias could be m dependent?

*3.* I think the introduction and related works are somewhat overlapped. The authors could provide more examples/applications in the introduction section to better motivate the problem set, at least the reasons for using m experts.

*4.* Typos: At the bottom of page 2, they "represented" a followed .... The citation of [Sadeghi & Fazel (2024)] is wrong; the paper was published in 2023.

----

---

### Meta-Review · Area_Chair_zLNC · 2024-12-20

**Metareview:**

This is an interesting paper on strategic combinatorial bandits.

There are one (actually two) main criticisms raised by the reviewers, and I can only agree with them.

The approximation ratio is certainly sub-optimal (1/2 vs 1-c/e) and the regret in T^{3/4} is also a-priori suboptimal (notice that if is possible to improve the approximation ratio, then the regret is actually linear in T).

I can see two alternatives to improve the paper
1. Get a better approx ratio, maybe in a more restrictive setting (as suggested by a reviewer).
2. Prove any type of lower bounds.

Notice that getting a \sqrt{T} regret without answering any of those two questions, might not be -- for me -- enough.

**Additional Comments On Reviewer Discussion:**

The reviewers agreed on the weaknesses of this paper, and I concurred with them on those points. It was a borderline paper, but it was at the end only slightly below the bar.

---

### Decision · Program_Chairs · 2025-01-22

Reject